Article 

# Generalisable 3D printing error detection and correction via multi-head neural networks

Douglas A. J. Brion ◉ [1] ✉ & Sebastian W. Pattinson ◉ [1] ✉

Material extrusion is the most widespread additive manufacturing method but its application in end-use products is limited by vulnerability to errors. Humans can detect errors but cannot provide continuous monitoring or real-time correction. Existing automated approaches are not generalisable across different parts, materials, and printing systems. We train a multi-head neural network using images automatically labelled by deviation from optimal printing parameters. The automation of data acquisition and labelling allows the generation of a large and varied extrusion 3D printing dataset, containing 1.2 million images from 192 different parts labelled with printing parameters. The thus trained neural network, alongside a control loop, enables real-time detection and rapid correction of diverse errors that is effective across many different 2D and 3D geometries, materials, printers, toolpaths, and even extrusion methods. We additionally create visualisations of the network's predictions to shed light on how it makes decisions.

Material extrusion is the most common additive manufacturing (AM) method for reasons including its relatively low-cost, little post-processing, compatibility with many materials and multi-material capability[1]. These have made extrusion AM promising in numerous areas[2] including healthcare[3], medical devices[4], aerospace[5], and robotics[6]. However, a key reason why many of these applications remain at the research stage is that extrusion AM is vulnerable to diverse production errors. These range from small-scale dimensional inaccuracies and mechanical weaknesses to total build failures[1,7–10]. To counteract errors, a skilled worker typically must observe the AM process, recognise an error, stop the print, remove the part, and then appropriately adjust the parameters for a new part. If a new material or printer is used, this process takes more time as the worker gains experience with the new setup[11,12]. Even then, errors may be missed, especially if the worker is not continuously observing each process. This can be difficult if multiple printers are in operation simultaneously or, as highlighted by the COVID-19 pandemic, personnel is limited due to social distancing or illness. Not only does this cost material, energy, and time but it also limits both the use of AM parts in end-use products, particularly safety-critical ones such as medical devices, and the resilience of AM-based supply chains. These challenges are set to become

more pressing as AM expands to living and functional materials, complex multi-material lattice structures, and challenging environments such as remote, outdoor construction sites or on the human body.

This has motivated diverse and interesting research into monitoring extrusion AM[13]. Current[14,15], inertial[16,17], and acoustic[18–22] sensors have often been used for monitoring extrusion AM. Although these approaches lead to the reliable detection of certain, typically large-scale, error modalities during printing, many errors remain undetectable. These methodologies are also yet to be used in most 3D printers, as the cost of sensors and amplifiers for such approaches is often high. Additionally, they are not sufficiently data-rich to enable online feedback and correction.

Camera-based approaches are potentially versatile and data-rich. Single cameras mounted on the printer frame with a top-down or side-on view, coupled with traditional computer vision and image processing techniques, have been used to detect diverse extrusion AM errors[23–32]. This approach has the advantages of being relatively inexpensive, easier to set up and that the camera can often view much of the manufactured part at any time. This allows many errors, such as infill deformation or the presence of material 'blobs', to be detected.

[1]Department of Engineering, University of Cambridge, Trumpington Street, Cambridge CB2 1PZ, UK. ✉e-mail: dajb3@cam.ac.uk; swp29@cam.ac.uk

However, using a single camera can limit the amount of information gained about the manufacturing process and thus the range of errors and error types identified. Multi-camera approaches are more expensive and complex to implement but potentially more capable. Multiple views of the part, or the addition of infra-red cameras, can allow defects, such as incomplete prints, to be seen that may not be apparent from a single viewpoint[33–35]. 3D reconstructions of printed parts, for example, generated by multi-camera 3D structured light scanning and digital image correlation, can be compared to the 3D digital part model to detect dimensional inaccuracies[35–43]. However, these more sophisticated systems are often expensive, sensitive to lighting conditions and part surface properties, slower due to scan time and computation, require precise positioning and calibration, and limited to detecting errors large enough to see given scanner resolution limits.

Frame-mounted single and multi-camera approaches, as above, also often find it difficult to view the material as it is being deposited from the nozzle because the print head can obscure the view. Typically, prints must be paused to allow imaging of a layer, which prevents real-time correction, slows production rates and can itself cause errors due to inconsistent extrusion. This has motivated work on mounting single and multiple cameras to the nozzle or extruder, which can view the ongoing printing process and has enabled real-time feedback to correct over or under extrusion during printing[44,45] as well as estimation of the shape of material extruded from the nozzle[46]. Traditional computer vision approaches are very promising for explicitly targeting specific errors in specific parts in 3D printing systems for which they have been calibrated. However, it is very challenging to handcraft feature extraction algorithms that can generalise to different parts, printers, materials, and setups. Hence, most examples only show a single combination of printer, part geometry, material, and printing condition, and none demonstrate correction of errors in multiple parts or setups.

Machine learning and particularly deep learning techniques have achieved state-of-the-art performance across many applications, including vision[47], by expressing complex representations in terms of other simpler representations[48]. This has led to several exciting recent demonstrations of machine learning in extrusion AM error detection[49–56]. Existing work has only demonstrated error detection in a single part, though, and thus the effectiveness of existing techniques for other parts, particularly parts not seen in the training data, is unknown. Moreover, most existing approaches can only detect a single error modality: poor flow rate[49], interlayer defects[50], warp deformation[53], and large top surface defects[52,54]. Often existing methods also require an object to already have been printed successfully to provide comparisons for the error detection[51,54,55]. This may be especially limiting for custom parts. Machine learning is most exciting in error detection because it could potentially be more robust and generalisable to new materials, geometries, and printers than hand-crafted features. However, the potential of machine learning algorithms to discover generalisable error features remains largely unexplored.

For error detection to reach its full potential in reducing 3D printing waste and improving sustainability, cost, and reliability, it must be coupled with error correction. There has been work in detecting and correcting some kinds of errors between subsequent prints of the same object[51,55]. However, many prints of that object are required to build the dataset, enabling error correction in that object. Moreover, these methods are not capable of real-time correction, meaning that if an error is detected, that part cannot be recovered. A previous study has examined real-time correction and control for extrusion AM[49]. However, the implementation only demonstrates correction of the flow rate printing parameter and only in one geometry that is used for both training and testing the system. There is also a significant delay between error detection and correction. As is the case with error detection, the performance of existing error correction methods in unseen objects is unclear, which limits their industrial applicability.

Here we report an easily deployable method using inexpensive webcams and a single multi-head deep convolutional neural network to augment any extrusion-based 3D printer with error detection, correction, and parameter discovery for new materials (Fig. 1). This has been realised in this work through the development of CAXTON: the collaborative autonomous extrusion network, which connects and controls learning 3D printers, allowing fleet data collection and collaborative end-to-end learning. Each printer in the network can continuously print and collect data, aided by a part removal system. Unlike existing deep learning AM monitoring work, which often uses the human labelling of errors to train algorithms, CAXTON automatically labels errors in terms of deviation from optimal printing parameters. Uniquely, CAXTON thus knows not just how to identify but also how to correct diverse errors because, for each image, it knows how far printing parameters are from their optimal values. This autonomous generation of training data enables the creation of larger and more diverse datasets resulting in better accuracies and generalisability. The final system is able to detect and correct multiple parameters simultaneously and in real-time. The multi-head neural network can self-learn the interplay between manufacturing parameters due to the single shared feature extraction backbone, even making the system capable of recognising multiple solutions to solve the same error. As part of this work, a large-scale, optical, in situ process monitoring dataset for extrusion AM has been curated and will be released. It contains over 1 million sample images of material deposition from the printer nozzle labelled with their respective printing parameters from 192 prints of different 2D and 3D geometries. The system is highly scalable, using commonly used firmware, and capable of growth through the remote addition of further printers for larger and more diverse future datasets. Despite being trained only on extruded thermoplastic polylactic acid parts, these capabilities generalise to previously unseen printers, camera positions, materials, and direct ink write extrusion. We also describe several innovations such as toolpath splitting and proportional parameter updates that enable correction speeds to be improved by an order of magnitude compared to currently published real-time 3D printing error correction work. This is enabled with low-cost equipment requiring only a network connection, a standard consumer webcam, and a low-cost single-board computer (e.g., Raspberry Pi). Finally, the use of attention layers within the network enables human operators to interpret what features the network focuses on. Visualisation methods are then employed to gain insights into how the trained neural network makes predictions both to aid fundamental understanding and to help build trust or enable traceability.

## Results

### Dataset generation, filtering and augmentation

We generated a new 3D printing dataset containing parts printed using polylactic acid (PLA), labelled with their associated printing parameters, for a wide range of geometries and colours using fused deposition modelling 3D printers. Our CAXTON data generation pipeline automates the entire process from STL file selection to toolpath planning, data collection and storage (Fig. 1a). Model geometries are automatically downloaded from the online repository, Thingiverse. The geometries are subsequently sliced with randomly sampled settings (e.g. scale, rotation, infill density, infill pattern and wall thickness). The generated toolpaths are then converted to have maximum moves of 2.5 mm by a custom Python script, to avoid excessively long moves executing a single parameter set and to reduce the firmware response time. During printing, images are captured every 0.4 seconds. Each captured image is timestamped and labelled with the current printing parameters: actual and target temperatures for the hotend and bed, flow rate, lateral speed, and Z offset. These parameters are precisely known by either retrieving values from firmware in real-time or by setting the value with a G-code command.

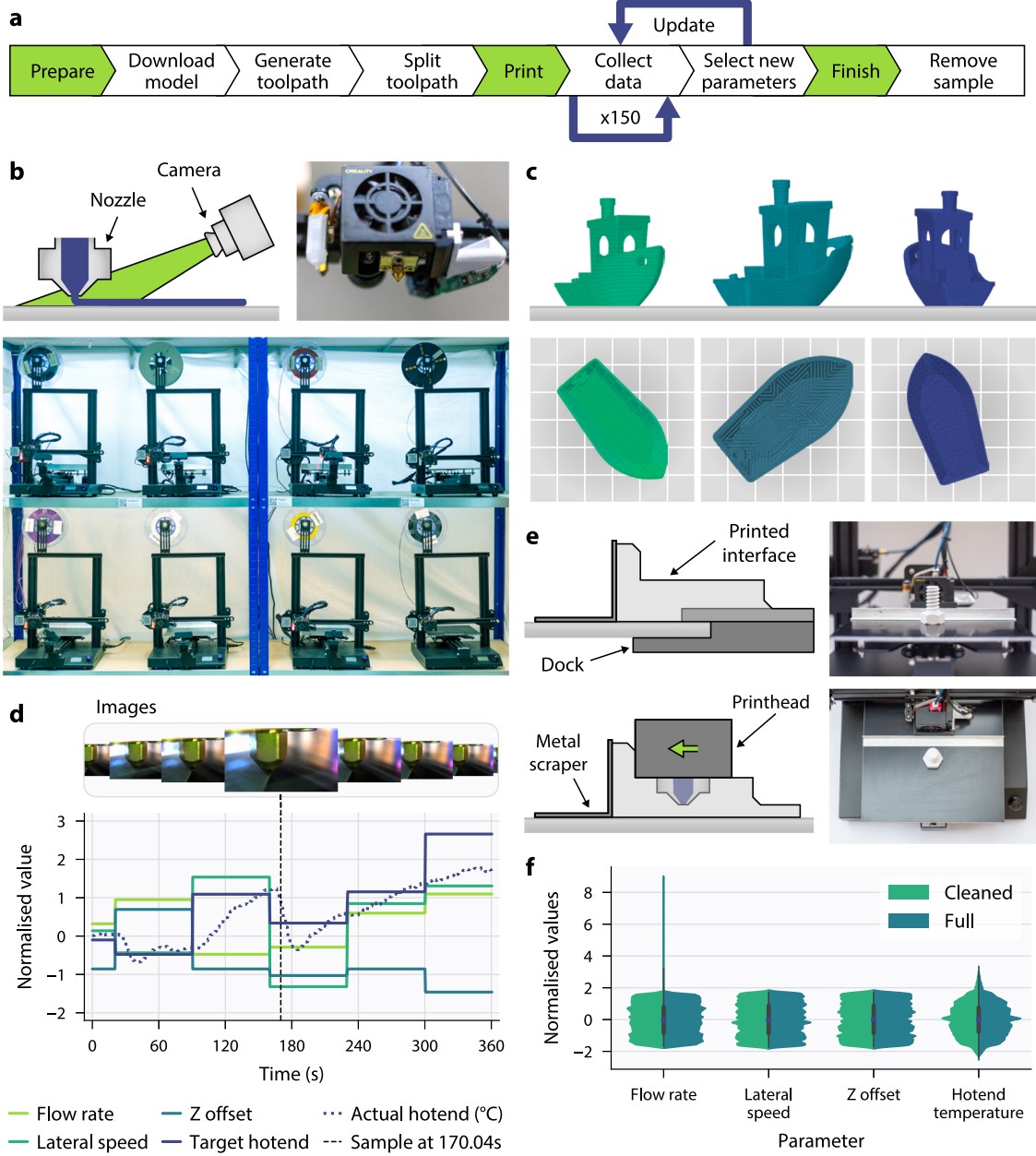

**Fig. 1 | Overview of the CAXTON system used for automated data collection.**
**a** Workflow for collecting varied datasets from extrusion 3D printers with the automatic labelling of images with printing parameters. **b** Fleet of eight thermoplastic extrusion 3D printers (Creality CR-20 Pro) equipped with cameras focused on the nozzle tip to monitor material deposition. **c** Renderings of generated toolpaths for a single input geometry, with randomly selected slicing parameters. **d** Snapshot of data gathered during an example print showing images with varying parameter combinations. **e** Design of bed remover and dock utilising existing motion system along with photographs taken during operation. **f** Distributions of normalised parameters in the full dataset collected by CAXTON containing over 1.2 million samples.

Additionally, for each image, nozzle tip coordinates on each printer are saved to allow for automated cropping around the region of interest during training. After 150 images have been collected, a new combination of printing parameters is generated for every printer by sampling uniform distributions of each parameter. The new parameter combinations are sent to each printer over the network as G-code commands which are subsequently executed with minimal delay due to the toolpath conversion. Upon execution, another 150 labelled images are gathered before the parameter update process happens again. This continues until the end of the print, and results in sets of images each with vastly different printing parameters (Fig. 1d). This automated labelling procedure for each image provides greater resolution than human-based labelling because no human operator could

label parameters with the same level of accuracy (for example, that the current flow rate is 56%), and no human could label an image with an exact combination of multiple printing parameters as they strongly interact with each other (for example, is the nozzle too high, flow rate too low, and temperature too low or a weighted combination of these).

Due to sampling suboptimal parameter combinations, some prints turn into complete failures, which after a certain point provide little information on the associated parameters. Such images are manually removed, leaving 1,166,552 labelled images (91.7% of the original 1,272,273). The remaining dataset contains some noisy labels due to the longer response times found when updating printing parameters, such as flow rate before a noticeable change is present in the image. The response time consists of a command execution delay

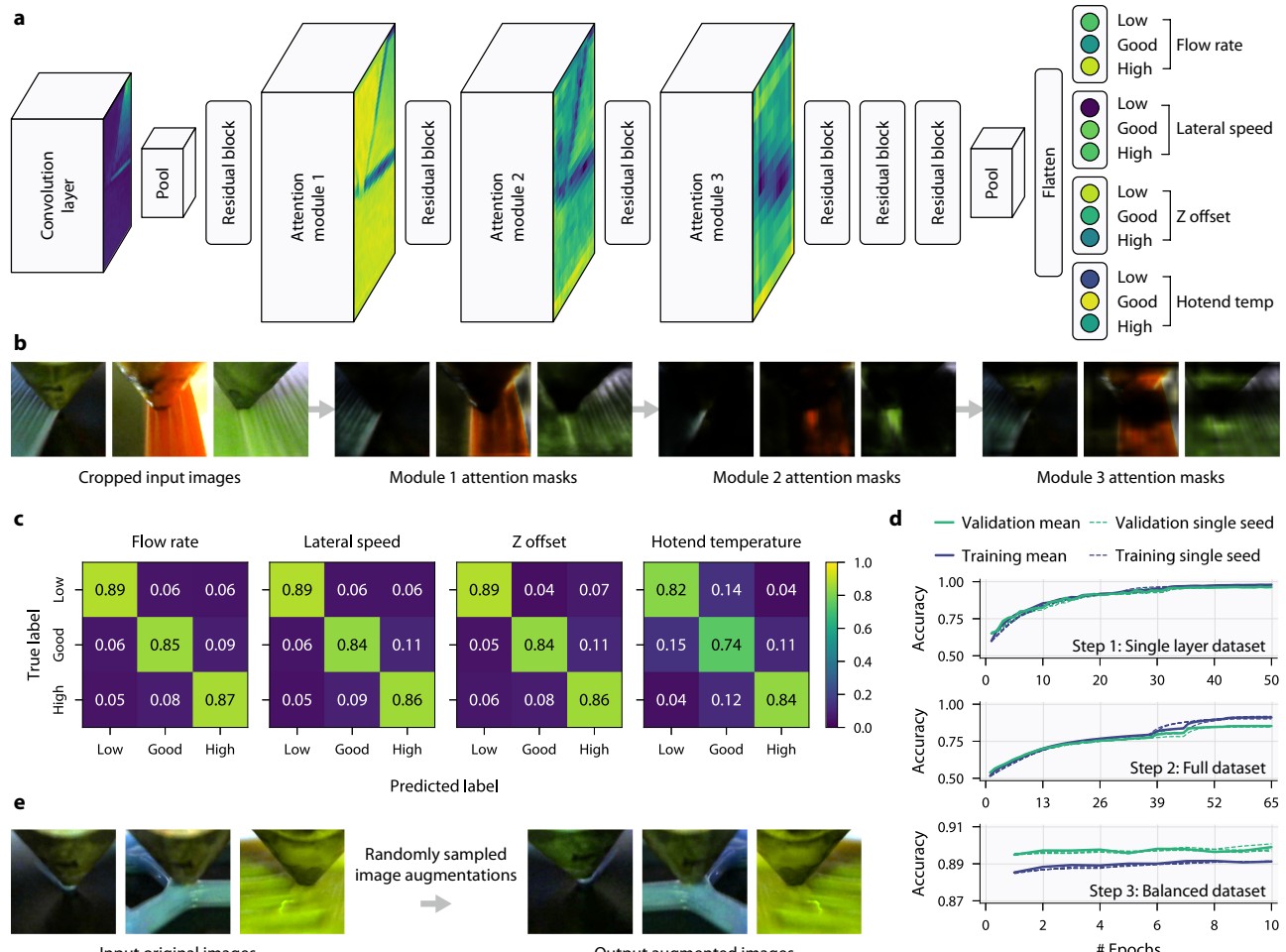

**Fig. 2 | Multi-head residual attention network architecture, performance, and visualisations for human interpretation. a** The multi-head network architecture consist of a single shared Attention-56 network[58] backbone, which contains stacked attention modules and residual blocks, followed by four separate fully connected output heads after the flattening layer, one for each parameter. Each of these heads classifies its associated parameter as either low, good, or high. Attention modules consist of a trunk branch containing residual blocks and a mask branch which performs down- and up-sampling. **b** Example attention masks at each module for the given input images. Each module output consists of many channels of masks, only a single sample is shown here. The masks show regions the network is focussing on, such as the most recent extrusion as shown by the output of module 2. **c** Confusion matrices of the final network after the three stages of training on our test dataset for each parameter. **d** Training and validation accuracy plots from training the network across three seeds, smoothed with an exponential moving average, on three datasets: single layer, full and balanced. **e** Example data augmentations used during training to make the model more generalisable.

and mechanical delay. The first delay is mostly handled by only capturing images after an acknowledgement of the parameter update command has been received from the printer. For mechanical delay, worst-case experiments were run to determine the response time for changing each parameter from the minimum to the maximum value in the dataset. It was found that changes are predominantly visible within 6 s of an update being applied, and as such, 15 images are removed post parameter updates. This leaves 1,072,500 samples where the system has reached its desired state. Unrealistic parameter outliers caused by printers not properly executing the G-code commands, or glitches in sensors such as thermistors are filtered, leaving 991,103 samples. Finally, very dark images with a mean pixel value across RGB channels of less than 10 are removed. This results in a cleaned dataset of 946,283 labelled images (74.4% of the original). The currently continuous parameter values are then binned into three categories for each parameter: low, good, and high. The upper and lower limits for these bins are selected based on our experience of AM with PLA. This creates a possible 81 different class combinations for the neural network to predict (three categories for four parameters).

We use data augmentation to increase the size and quality of our filtered dataset and thus avoid overfitting and improve generalisability

in our model[57]. The location and shape of the deposited material in the captured images varies greatly depending upon the geometry of the part being printed. Additionally, it was found that colour, reflectance, and shadows all differed with camera position, material choice and printer design. To simulate a wider variety of geometries, camera locations and materials, each image in the dataset is subjected to a wide range of data augmentation techniques (Fig. 2e). First, the full-sized image captured by the camera is randomly rotated by up to 10 degrees in either direction. Then a minor perspective transform is applied with a probability of 0.1. The image is then automatically cropped to a 320 × 320 pixel square region focused on the nozzle tip using the nozzle tip coordinates saved during data collection. The rotation and perspective transforms are applied before the crop to practically remove the need for padding in the cropped region. A random square portion with an area between 0.9–1.0 of the 320 × 320 image is then cropped and resized to 224 × 224 pixels—the input size for the deep neural network. Subsequently, a horizontal flip can be applied to the image with a probability of 0.5, followed by applying a colour jitter of ±10% to the image's brightness, contrast, hue, and saturation. This use of synthetic data augmentation is more time and resource efficient than repositioning cameras on printers and

changing environmental lighting conditions during the dataset collection. It also allows for a smaller raw dataset with augmentations functionally applied at run-time during training instead of increasing the dataset size with more samples. Finally, the channels in the transformed image are normalised using each channel's pixel mean and standard deviation for all the images in the filtered dataset.

## Model architecture, training and performance

The accurate prediction of current printing parameters in the extrusion process from an input image is achieved using a multi-head deep residual attention network[58] with a single backbone and four output heads, one for each parameter. In deep learning, single-label classification is very common and requires only a single output head to classify the input as one of $N$ possible classes. However, this work requires multi-label classification to classify the input as one of three possible classes (low, good, and high) for each of the four labels (flow rate, lateral speed, Z offset and hotend temperature). To achieve this multiple output heads are used with a shared backbone for feature extraction. The weights of the shared backbone are updated during the backward pass in training by a sum of the losses from each of the separate output heads. This allows the backbone to learn its own interpretation of the relationships between each of the parameters and the importance of certain features shared across the parameters. The alternative approach is to use multiple separate networks, each with a single output head and to treat the problem as four separate single-label classification problems. This, however, looks at each parameter in isolation and, as such, fails to learn the interplay and relationships. Additionally, it requires significantly more compute during training and in real-world deployment as four separate networks must be trained independently (as opposed to one), and then these networks must be run in parallel during operation.

The use of attention in the network may reduce the number of network parameters needed to achieve the same performance for our application, whilst making the network more robust to noisy labels. The attention maps may also aid in inspecting errors and explaining predictions. The single backbone allows for feature extraction to be shared for each parameter and, as such reduces inference time compared to having separate networks. Additionally, it allows the single network to model the interplay between different parameters. Each head has three output neurons for classifying a parameter as low, good, or high. With this structure, the network predicts the state of the flow rate, lateral speed, Z offset, and hotend temperature simultaneously in one forward pass from a single RGB input image (Fig. 2a). This multi-head structure and knowledge of multiple parameters may lead to an improvement in predicting individual parameters. Interestingly, it was found that a network solely trained to predict flow rate class achieved a lower accuracy in classifying flow rate than a network trained with knowledge of all four parameters. However, further experiments are required to examine this result and whether additional context can be used to increase network performance.

The shared network backbone consists of three attention modules and six residual blocks and is based on the Attention-56 model[58]. The attention modules are composed of two branches: the mask and the trunk. The trunk branch performs the feature processing of a traditional network and is constructed from residual blocks. The mask branch undertakes down sampling followed by upsampling to learn an attention mask with which to weight the output features of the module. This mask can not only be used during the forward pass for inference, but also as a mask in the backward pass during backpropagation. This was one of the reasons for choosing this network architecture, as it is believed that these mask branches can make the network more robust to noisy labels—which our dataset contains due to parameter changes and subtle inconsistencies during printing. After these blocks, the network backbone is flattened to a fully connected layer which links to each of the separate four heads. The

heads need to be separate outputs of the network this work requires multi-label classification as each full prediction requires each head to always have a separate single prediction. An alternative approach would be to use four separate complete neural networks; however, this would be significantly more compute and memory intensive in addition to not being able to model the relationships between manufacturing parameters. The multi-head shared backbone approach used in this work results in the backbone being used as a feature extractor to compress the dimensionality of the input image into a latent space representation learned from the sum of losses for each manufacturing parameter. It can then be thought that each head acts as a mapping from this latent space to the classification of the parameter level.

To visualise which features the network is focussing on at each stage, images of the attention maps after each module were created (Fig. 2b). Here, the same attention mask from each module is applied to each of the 3 input images with the areas not of interest darkened (note: these masks are illustrative examples as each module contains many different attention maps). The network appears to focus on the printed regions in the example mask output for attention module 1, and then only on the most recent extrusion for module 2. Module 3 applies the inverse to the previous, focusing on everything but the nozzle tip.

It was found that splitting the training process into three separate stages and using transfer learning was the most robust. For each stage three differently seeded networks were trained. In the first stage, the network is trained on a sub-dataset containing only images of the first layers with 100% infill. The features are more visible for each parameter in these prints and by first training, with this subset, the network can more quickly learn to detect important features. It was found that this separation sped up the learning process as features were more learnable for the single layer and could subsequently be tuned on the full dataset making the network generalisable to complex 3D geometries. A training accuracy of 98.1% and validation accuracy of 96.6% was achieved by the best seed. A transfer learning approach was then used to retrain the model of the best seed on the full dataset containing images for all 3D geometries. This was done three times, with the best seed achieving a training and validation accuracy of 91.1 and 85.4%, respectively. Neural networks can learn inherent biases in the data given to them; therefore, due to imbalances in our full dataset (for example, the Z offset can have many more values which are too high than too low because the nozzle would crash into the print bed) transfer learning was used a final time. This time, however, only the final fully connected layer to each of the four heads was trained on a balanced sub-dataset containing an equal number of samples for each of the possible 81 combinations (four parameters, each of which can be low, good or high). The weights in the network backbone for feature extraction were frozen. This achieved a training accuracy of 89.2% and validation of 90.2%. Then, the final trained network was tested on our test set, where it achieved an overall accuracy of 84.3%. For each parameter, the classification accuracies on our test set were: flow rate 87.1%, lateral speed 86.4%, Z offset 85.5% and hotend temperature 78.3%. More information on the training process can be found in Supplementary Fig. S1. Not only would this task be very challenging for an expert human operator, given the diverse, multi-layer test set, but this accuracy also understates the efficacy of the network for error correction. The parameters are interdependent and thus, for many error types, there will be multiple combinations of parameter changes that could correct it. For example, a higher Z offset with the nozzle far from the print bed can easily be mistaken as having a low flow rate—both appear as under extrusion—and could be corrected by changing either parameter. However, only one of these parameter combinations will be counted as 'correct' in the accuracy calculation given the labels in the training data.

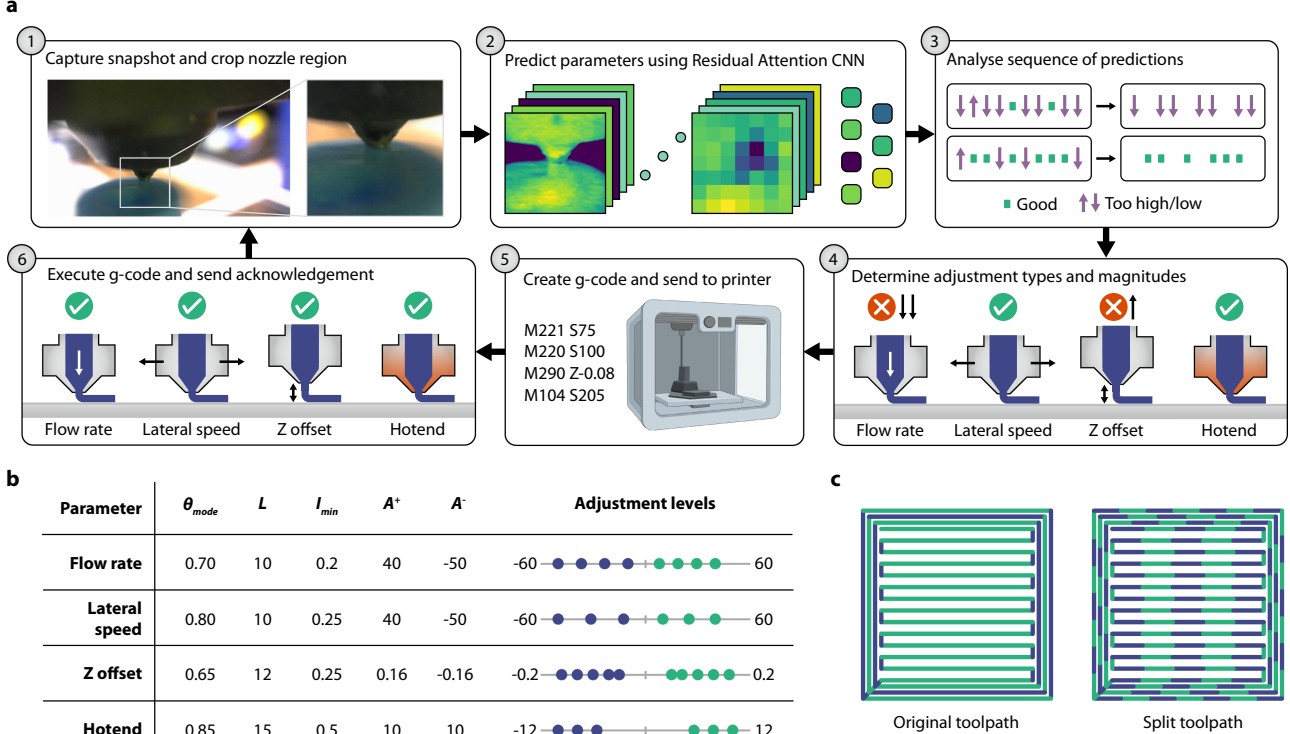

**Fig. 3 | Machine vision control system pipeline and feedback parameters. a** The six major steps in the feedback pipeline enable online parameter updates from images of the extrusion process. **b** Table containing $\theta_{mode}$ (mode threshold), $L$ (sequence length), $I_{min}$ (interpolation minimum), $A^+$ (the largest increase), $A^-$ (largest decrease) for each printing parameter along with the possible levels of update amounts. **c** Simple example single layer geometry illustrating toolpath splitting into equal smaller segments. 1 mm lengths are used in the feedback process to enable rapid correction and reduce response time.

## Online correction and parameter discovery pipeline

To test the ability of the network to correct printing errors and discover optimal parameters for new materials, random 3D models were again downloaded, but this time for testing correction. Each 3D model was sliced with different settings for scale, rotation, infill density, number of perimeters and number of solid layers by randomly sampling from uniform distributions. The infill pattern was chosen at random from a given list of common patterns. The set of toolpaths generated were subsequently converted to have maximum moves of 1 mm using a custom script to enable significantly faster firmware response times for parameter changes during printing while keeping print file sizes manageable and preventing jitters due to the printer not being able to read and process the G-code lines quickly enough.

During the printing process, images of the nozzle tip and material deposition are taken at 2.5 Hz and sent to a local server for inference (Fig. 3a). Each received image is automatically cropped to a 320 × 320 pixel region focused on the nozzle tip. The user needs to specify the pixel coordinates of the nozzle once when mounting the camera at setup. Furthermore, users may want to alter the size of the cropped region depending on the camera position, focal length, and size of the printer nozzle. Choosing a suitable region around the nozzle affects the performance of the network and the best balance between accuracy and response time is seen when ~5 extrusion widths are visible on either side of the nozzle tip.

The cropped image is then resized to 224 × 224 pixels and normalised across RGB channels. Next, the classification network produces a prediction (too high, too low, good) for each parameter given this image as input. These predicted parameters are stored in separate lists of different set lengths, $L$, for each parameter. If a particular prediction is made frequently enough that it makes up a proportion of a full list greater than or equal to the mode threshold ($\theta_{mode}$), then a mode is found, and that prediction is accepted. If no mode is found,

then no updates are made, and the printing parameter is considered acceptable, just as in the case where the mode prediction is 'good'. If a mode is found to be 'too high' or 'too low', the proportion of the list length constituted by the mode value is used to scale the adjustment to the parameter facilitating proportional correction. Specifically, one-dimensional linear interpolation is applied to map the range between a parameter threshold ($\theta_{mode}$) and 1 to a new minimum ($I_{min}$) and 1. The interpolated value is then used to linearly scale the maximum update amount ($A^+$ for parameter increases and $A^-$ for decreases). The specific values of $\theta_{mode}$, $L$, $I_{min}$, $A^+$ and $A^-$ were obtained iteratively via experimentation for each parameter individually (Fig. 3b) to balance response time with accuracy and to prevent overshooting. The hotend list length and mode threshold are particularly conservative due to the long response time of this parameter and also the safety risk in overshooting.

Once the final update amounts have been calculated for the printing parameters, they are sent to a Raspberry Pi attached to each printer. The Pi retrieves the current value for each parameter and creates a G-code command to update the parameter. The Pi then looks for acknowledgement of the command's execution by the firmware over serial. Once all commands have been executed by the firmware, the Pi sends an acknowledgement to the server. When the server receives an acknowledgement that all updates have been executed it begins to make predictions again. Waiting for this acknowledgement of all parameter updates is crucial to stop oscillations caused by over and undershooting the target.

To demonstrate the system's correction capability, an experimentation pipeline was constructed to take an input STL file, slice it with good print settings, insert a G-code command to alter a parameter to a poor value and then parse the generated G-code and split the model into 1 mm sections (Fig. 4). The same model of the printer was used as in training but with an altered camera position (slightly rotated

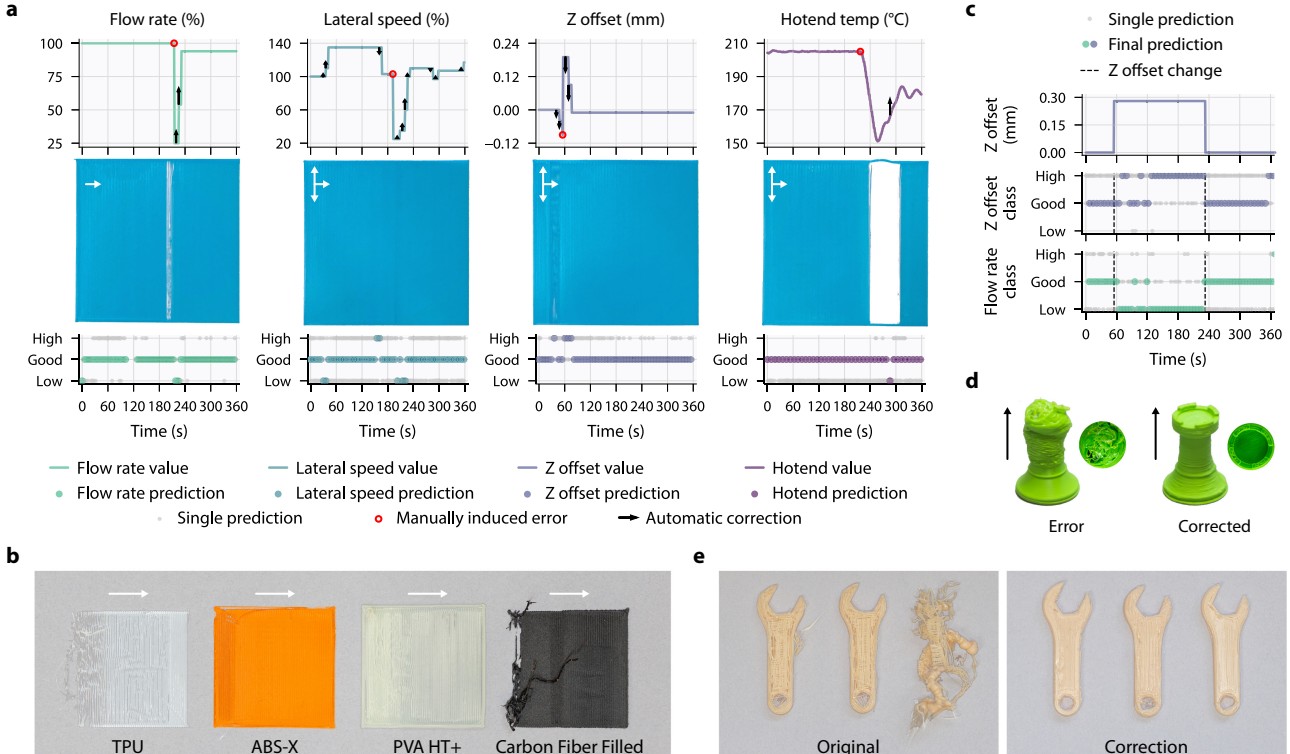

**Fig. 4 | Printer and feedstock agnostic online parameter correction and discovery. a** Rapid correction of a manually induced erroneous single parameter using the trained multi-head neural network. Printed with PLA feedstock on a known printer with an unseen 0.4 mm nozzle not used in training data. **b** Online simultaneous optimisation of multiple incorrect parameters on unseen thermoplastic polymers. Demonstrates that the control pipeline is robust to a wide range of feedstocks with different material properties, colour and initial conditions. **c** Much

like a human operator, the system uses self-learned parameter relationships for corrective predictions. A high Z offset can be fixed by both reducing the Z offset and/or by increasing the material flow rate. **d** Correction of multiple incorrect printing parameters introduced mid-print. Both rooks were printed in the same conditions, with the only difference being correction. **e** Correction of prints started with incorrect parameter combinations. All six spanners were printed with the same conditions.

and translated with respect to the nozzle), a new 0.4 mm nozzle with different external geometry, and an unseen single layer printing sample. To compare the responses between parameters, each was printed using the same spool of PLA filament (Fig. 4a). These single-layer prints are used as a clearly interpretable benchmark to test each of the individual parameters and combinations of parameters across different printers, setups, and materials. The flow rate, Z offset, and hotend temperature parameter defects are clearly visible, while the lateral speed defect can be observed as a darker line where print speed was slowed. The delay between the command being sent (black arrows in Fig. 4a) and the parameter updating is observable, demonstrating the importance of waiting for acknowledgements from the printer. In each case, the network, in combination with mode thresholding, is rapidly able to recover good printing parameters (see Supplementary Movie S1).

Despite being trained only using extruded thermoplastic PLA, the control pipeline generalises to diverse materials, colours, and setups. Figure 4b shows online correction for four different thermoplastics printed with different combinations of random multiple incorrect printing parameters on similar interpretable single layer benchmarks as Fig. 4a. In each case, the network successfully updates multiple parameters resulting in good extrusion (see Supplementary Movie S2). The TPU and carbon fibre-filled samples have no printed perimeter due to poor initial conditions. Not only is this useful for automated parameter discovery, aiding users in tuning their printers for new materials by quickly obtaining the best parameter combinations, but also it shows that control systems can improve productivity by saving failing prints where the initial toolpaths fail to adhere to the bed.

Thanks to having all parameter predictions in one network structure, the trained model learns the interactions between multiple parameters and can offer creative solutions to incorrect parameters like a human operator. We printed a sample using the control loop setup but without making online corrections. This sample contained a region with a high Z offset. A high Z offset results in separated paths of extruded material—the same result can occur from a low flow rate. Figure 4c shows that the network determines that increasing the flow rate along with lowering the Z will result in good extrusion. As the trained model can find multiple ways to solve the same problem, it can be more robust to incorrect predictions for a single parameter and enable faster feedback by combining updates across multiple parameters. The prediction plots also demonstrate the speed at which the network notices that parameters are now good, which is vital to ensure the control system does not overshoot when making online corrections.

Figure 4d applies the control pipeline using the same printer model as used in training (Creality CR-20 Pro) on an unseen rook geometry to demonstrate that our methodology could be used in a production setting for full 3D geometries. Multiple random incorrect printing parameters were introduced halfway through printing, specifically a very high flow rate, lateral speed and hotend temperature and a low Z offset. The rook printed without correction dramatically failed, whereas the rook printed with the same conditions with correction enabled was completed successfully. Figure 4e shows six copies of the same 3D spanner geometry, each started with the same combination of incorrect printing parameters: low flow rate and lateral speed, high Z offset and good hotend temperature. Of the six spanners, three were printed without correction resulting in one complete

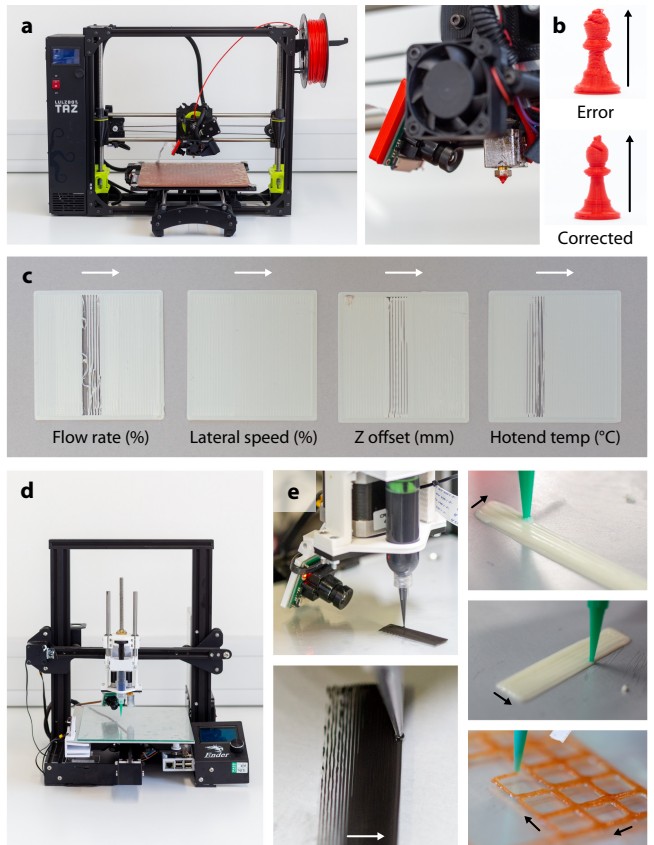

**Fig. 5 | Approach generalises across setups and extrusion processes. a** Photos of a tested unseen 3D printer with a 0.6 mm inner diameter nozzle (Lulzbot Taz 6). A different camera model (Raspberry Pi Camera v1) and lens were used compared to the collection of training data along with a new camera position with respect to material deposition. **b** A bishop chess piece with erroneous parameters introduced and the same erroneous print with correction enabled. Both were printed using 2.85 mm PLA on the unseen Lulzbot Taz 6 setup. **c** Rapid correction of a manually induced erroneous single parameter using the trained multi-head neural network. Printed with white PLA feedstock on an unseen printer with an unseen 0.6 mm nozzle not used in training data. **d** Syringe-based printer for direct ink writing (DIW) modified from a Creality Ender 3 Pro. An unseen camera model (Raspberry Pi Camera v1) and lens were used along with a different camera position. **e** Automated correction and parameter discovery showed for PDMS with 0.21 mm nozzle (27-gauge), alongside mayonnaise and ketchup with 0.84 mm nozzle (18-gauge).

failure due to detachment from the print bed and a very poor surface finish on the remaining two. These errors are due to the poor initial layer caused by the suboptimal printing parameters. The three printed with correction were all completed successfully and exhibit the same improved surface finish, particularly on the initial layer. It should be noted that these corrected prints do not match a perfectly printed part. Imperfections are present until all the necessary corrections have been applied, and as such, some of the initial layer is printed with poor starting parameters. Though rare, a correction can also be applied when it is not needed leading to an imperfection.

To demonstrate the system's generality, a different camera and lens were attached to a new location on an unseen printer (Lulzbot Taz 6) with a differently shaped nozzle and nozzle width—0.6 mm instead of 0.4 mm as used in training (Fig. 5a). This printer uses an extrusion system which takes 2.85 mm diameter filament as input over 1.75 mm as used in the training printers. Figure 5b shows the same control system applied to an unseen bishop geometry. Random incorrect printing parameters were introduced early on in the print, specifically during layer 7. These incorrect parameters were a low lateral speed and high flow rate, Z offset and hotend temperature. The erroneous bishop

printed without correction failed, whereas the bishop printed with the exact same conditions with the control pipeline enabled was completed successfully with greater detail. Single-layer benchmark prints were completed with each individual erroneous parameter introduced using white PLA (Fig. 5c). These demonstrate that the multi-head neural network and control pipeline generalise to correct parameters across fused deposition modelling printers. The size of the poorly printed region in these samples appears larger than in the ones printed for Fig. 4a, as the larger nozzle on the Lulzbot Taz six results in a far larger extrusion width for each line. The number of lines is approximately the same between the printers.

The control pipeline was further tested on a direct ink writing setup using a stepper motor with a threaded rod to move a plunger in a syringe (Fig. 5d). This used a different camera model and lens mounted at a different angle and distance from the nozzle with a transparent and reflective glass print bed instead of the black bed used during the thermoplastic tests. With this setup, PDMS, mayonnaise and ketchup were printed using a variety of nozzles −0.21 mm for the PDMS and 0.84 mm for the condiments (Fig. 5e). All samples were printed at room temperature with no hotend correction. For PDMS printing, the network only corrected the flow rate. Figure 5e shows that for PDMS, the network learns to increase flow rate by raising the pressure applied to the syringe. Once the required pressure is reached, the network reduces the flow rate to stop over extrusion. However, during long prints, the flow rate sometimes overshoots due to a large build of pressure in the syringe, especially when the network does not reduce the flow rate fast enough. Balancing this pressure is especially challenging in this specific setup due to the viscous material and small nozzle diameter requiring high pressures for printing, creating a time gap between plunger movement and extrusion. When printing less viscous materials this overshoot and pressure delay are less of a problem, especially with larger nozzle diameters. For the mayonnaise and ketchup examples, the network mostly adjusted the flow rate and Z offset. We found both condiments tended to over extrude, and the network often reduced the flow rate and, for the first layer, lowered the Z offset. When printing multi-layered structures, the network tended to raise the Z offset at each layer and reduce the flow rate to stop the nozzle tip from being submerged in the previous layer.

## Gradient-based visual explanations of network predictions

It is helpful to seek possible explanations for why models make certain decisions, particularly when deploying deep neural networks in production for safety-critical applications. Two popular visualisation methods that may help users gain some understanding of why neural networks make their predictions are guided backpropagation[59] and Gradient-weighted Class Activation Mapping (GradCAM)[60]. The former helps to show finer resolution features learned by the network in making predictions and the latter provides a coarser localisation showing important regions in the image (this can be thought of as post hoc attention). For both approaches, the target category (low, good, and high) for each of the four parameters is provided to determine which features or regions are specifically important for that category. On top of this, a method was developed to apply the techniques for each parameter separately within the whole network allowing us to produce up to 12 tailored visualisations for an input image (the three classes for each of the four parameters, e.g. low flow rate, high lateral speed, good Z offset).

Multiple combinations of erroneous parameters can result in either separated paths of extruded material (under extrusion) or overlapping paths of material (over extrusion). Guided backpropagation was used to try to determine if the network uses similar features across parameters to detect these physical extrusion properties. Representative example images for under, good, and over extrusion caused by different parameters are shown in Fig. 6a. It appears that similar features are shared between parameters for the

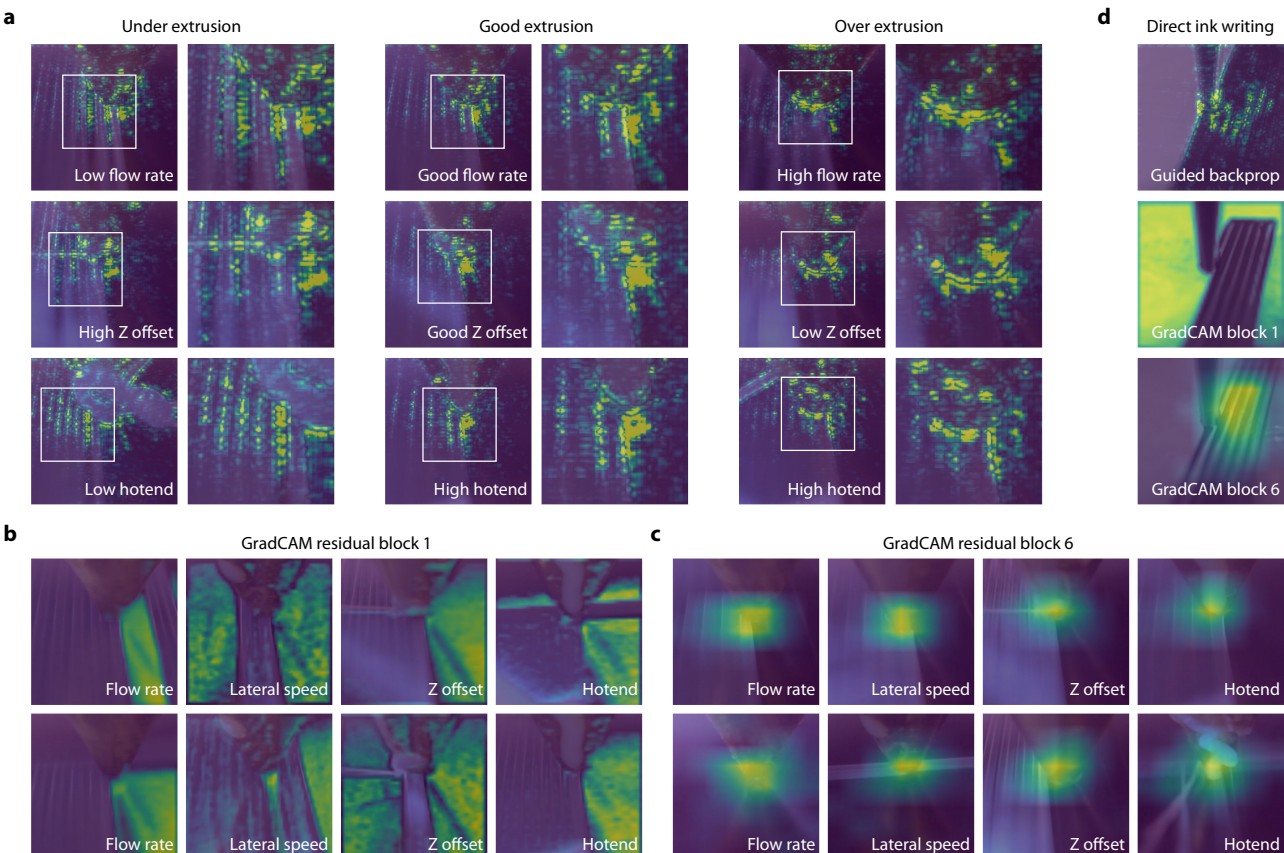

**Fig. 6 | Visual explanations using separate saliency maps for each parameter may assist in verifying the robustness of the network. a** Under, good or over extrusion can be achieved by multiple incorrect parameters. Guided backpropagation[59] is applied to highlight important features in the image used for classification. Representative example unseen images suggest that the network uses similar features across parameters to identify the same physical property. **b** Gradient-weighted Class Activation Mapping (GradCAM)[60] shows that across parameters and unseen inputs, the early stages of the network differentiate between the deposited material and the print bed. **c** GradCAM applied to the final stages shows that the network as a whole focuses on the nozzle tip across parameters and unseen inputs. **d** The trends shown in guided backpropagation and GradCAM at different stages also apply to different unseen extrusion methodologies, such as direct ink writing.

same extrusion classification: separated paths for under extrusion, an outline of the current path for good extrusion, and around the nozzle for over extrusion.

GradCAM was applied to every layer of the shared network backbone for each of the parameters separately. We show in Fig. 6b, c the visualisations from the first and last layers (residual blocks 1 and 6, respectively). Earlier stages in the network appear to detect large structural features in the image, such as differentiating between the deposited material and the print bed. By the last layer, the network predominantly focuses on the most recent extrusion from the nozzle irrespective of parameter or target class. This is desired as for fast response times and corrections, we want the network to use information from the most recently deposited material for its prediction. In Fig. 6d, example visualisations are shown images from direct ink writing tests. These images demonstrate that the trained network can use similar features at each stage during prediction as it uses for thermoplastic predictions. Further visualisations can be found in Supplementary Fig. S2 and Supplementary Movie S3.

## Discussion
We demonstrate that training a multi-head neural network using images labelled in terms of deviation from optimal printing parameters enables robust and generalisable, real-time extrusion AM error detection and rapid correction. The automation of both data acquisition and labelling allows the generation of a training image-based dataset sufficiently large and diverse to enable error detection and

correction that generalises across realistic 2D and 3D geometries, materials, printers, toolpaths, and even extrusion methods. The deep multi-head neural network was able to simultaneously predict with high accuracy the four key printing parameters: flow rate, lateral speed, Z offset and hotend temperature from images of the nozzle during printing. It was found that this additional context and knowledge of multiple parameters even may lead to an improvement in predicting individual parameters—though further research to support this finding is needed. Like a human, the system was able to creatively propose multiple solutions to an error and could even discover new parameter combinations and learn how to print new materials. Unlike humans, though, the system operated continuously and made corrections instantaneously. Alongside this network, we present numerous advances in the feedback control loop with new additions such as proportional parameter updates, toolpath splitting, and optimised prediction thresholding, which combined provide an order of magnitude improvement in correction speed and response time compared to previous work.

Whilst significantly advancing the capabilities of AM feedback through the generalised control of more parameters with faster response times, this work also lowers the cost and complexity of existing approaches. The use of off-the-shelf cameras, small single-board computers (e.g., Raspberry Pi), and networking allow the system to be added to new and existing printers with ease. The system extends and plugs into popular software and firmware packages meaning that existing user workflows are minimally affected. Furthermore, the fully

in-built networking enables each added printer to increase the training data available and also allows for the system to be deployed in more remote environments where only an internet connection is required.

There is room for improvement in the methodology. For example, testing the network on a wider range of printers and materials and adding data gathered to the training dataset could make the system more generalisable and robust. More data for low Z offset values may also be beneficial as small differences in value can have a large impact on print quality. Also, there is a smaller range of values low Z offsets can take over high ones before hitting the bed, this causes a bias in the dataset as more values are present in the high classification. Plus, we believe the small movements in Z offset may be the primary weakness of the current dataset and improved camera focus, resolution and attention to positioning would greatly improve the next iteration of large AM datasets. It is also important to raise the role which bias may play in the performance of the trained model given the dataset provided. Future work would be enhanced by using an even larger and more balanced dataset with an equal number of samples at more granular levels of classification. For example, at present, in the dataset, there may be certain combinations of parameters which only appear in during a specific print or with a single colour of filament, and thus the network has learned these incorrect features as mappings. Additionally, while 3D models, slicing settings, and parameter values have been randomly sampled, there is still some bias including in the given ranges, additional slicing settings, and feedstock choice.

The effectiveness of our methodology can be further improved by tuning the many variables used during the online correction feedback pipeline, along with the sampling rate and toolpath split length. More extensive testing with a wider and deeper search of values may yield enhanced performance. Better values for these correction variables would help to reduce the chance of correction oscillations during feedback which we have experienced in testing by improving predictions over lists or reducing response time. Parameter oscillations can occur if the network can still see a previously bad region and overshoots its correction or upon a series of incorrect predictions from the neural network.

Furthermore, we realise that whilst this approach helps solve many common extrusion-based printing errors, many still remain. Mechanical failures on the printer caused by skipped steps, belt slipping, or external interference remains unsolved and the addition of closed-loop control for positioning errors would increase the number of error modalities covered. Electrical issues caused by faulty sensors or underperforming power supplies may be detectable in some instances but cannot be autonomously corrected. Additionally, large errors such as cracking, warp deformation, and bed adhesion issues resulting in part detachment are not entirely solved. Whilst the precise real-time control of printing parameters can help reduce the likelihood of these errors occurring, it is not able to detect or resolve many of them once they have formed due to its localised approach to monitoring. Combining this local imaging with a global camera system may yield significant improvements in detecting more errors and could provide a link between local extrusion issues and global scale faults.

The gradient-based saliency maps we used to examine how the network reaches its decisions suggest that the network learns to focus on the most recent extrusion when making predictions, which aids rapid response to errors. This, together with the ability of the network to accurately predict different parameters across different geometries, materials and setups suggest that the network identifies visual features that are universal to extrusion processes, such as the shape of the extrudate. The methodology developed in this paper is, to a large extent, agnostic to the sensors and manufacturing process it is applied to. This points to a range of areas for future investigation. For instance, in integrating new infra-red or other sensors into the system or applying it to further challenging manufacturing processes[61,62]. Applying it to metal AM methods is particularly exciting given the complexity of these processes and the need for quality assurance[63]. Optical techniques are the most common methods used thus far in metal AM for monitoring features such as the powder bed surface and melt pool[64,65]. These would be appropriate for use with our methodology and may be especially beneficial for AM of metals that it can be difficult to work with[66]. This could be aided by fine-tuning the model on specific setups with transfer learning on small, specialised datasets (with uniformly good lighting), to boost performance in known environments.

## Methods

### CAXTON system for autonomous data collection

A network of eight FDM 3D printers were used for data collection. Creality CR-20 Pro printers were chosen due to their low cost, pre-installed bootloader and included Z probe. The firmware for each printer was flashed to Marlin 1.1.9 to ensure thermal runaway protection was enabled. Each printer was equipped then with a Raspberry Pi 4 Model B acting as the networked gateway for sending/receiving data to/from the printer via serial. The Pi runs a Raspbian-based distribution of Linux and an OctoPrint server with a custom-developed plugin. A low-cost, consumer USB webcam (Logitech C270) was connected to the Pi for taking snapshots. The camera was mounted facing the nozzle tip using a single 3D printed part. These components can easily be fitted to new and existing printers at low cost; aiding calability and deployability.

The printer used for direct ink writing was a modified Creality Ender 3 Pro. The extruder setup was designed and built in-house and utilised a stepper motor-driven syringe with a Luer lock nozzle. The printer is equipped with a Raspberry Pi 4 Model, Z probe and Raspberry Pi Camera v1 with a zoom lens. The firmware is a configured version of Marlin 2.0. For further experiments, a Lulzbot Taz 6 with its firmware flashed to Marlin 1.1.9 was used. The default nozzle was swapped for an E3D 0.6 mm inner diameter nozzle.

STL files were downloaded from the Thingiverse 3D model repository using a Python script. This tool allowed models in the repository to be easily searched for by multiple avenues, such as keyword, popularity, time, creator, and license. With this tool, popular files (to increase the likelihood that they were printable) with appropriate licences were pseudo-randomly selected and downloaded. Additionally, some of the standard 3D printing benchmark files were manually added to this set of STLs.

For slicing parts to create varied toolpaths, STLs were randomly rotated (angle sampled from a uniform distribution ranging from 0° to 360°) and scaled before being centred on the build plate (scale factor sampled from a uniform distribution ranging from 0.8 to 2—for some already large models the scale factor was clipped to reduce printing time). Then the number of solid top and bottom layers was randomly sampled from a uniform distribution ranging from 2 to 4 along with the infill pattern, infill density (0% to 40%), and the number of external perimeter walls (2 to 4). The following range of infill patterns were used: rectilinear, grid, triangles, stars, cubic, line, concentric, honeycomb, 3D honeycomb, gyroid, Hilbert curve, Archimedean chords, and octagram spiral. The lines in the G-code output from the slicer were subsequently chopped into smaller segments with a maximum move length of 2.5 mm to reduce the firmware response times For the online correction demonstrations with a maximum move length of 1 mm used instead to further reduce firmware response time.

During printing, images are captured at a resolution of 1280 × 720 pixels from the nozzle-facing camera at a sampling rate of 2.5 Hz. Each image is labelled with the actual and target hotend and bed temperatures at that point in time and the printer's current relative flow rate and lateral speed (both percentages) along with the Z offset (in mm). After 150 of these labelled images have been collected and stored (~1 min of printing), new flow rate, lateral speed, Z offset, and hotend target temperature values are randomly sampled from uniform

**Table 1 | Comparison of model size and types and their respective test set accuracies across all four parameters**

| Model type | Test accuracy across four parameters |
|---|---|
| ResNet18 | 80.4% |
| ResNet34 | 82.5% |
| ResNet50 | 81.8% |
| ResNet101 | 81.4% |
| This work | 84.3% |

distributions of the following respective ranges of flow rate: 20 to 200%, lateral speed: 20 to 200%, Z offset: −0.08 to 0.32 mm, and hotend: 180 to 230 °C. It was found that some additional flow rates at higher levels needed to be added to the training set as they were sufficiently out of distribution that the trained models would incorrectly predict the classification. This was not necessary for the other parameters. After sampling, the new values are sent to the printer. The printer begins capturing another 150 images for this new combination of parameters. This process can happen in parallel across all eight printers we used, each of which uses a different colour of feedstock, helping to cover the large parameter space.

Random parameter value selection was chosen over systematic parameter selection to provide different surrounding contexts in the captured images. Specifically, by choosing a randomised approach, the outer region of the image can contain extrusion for a significantly different previous parameter combination, and this may help train the network to use local features around the nozzle tip. A systematic approach may instead introduce patterns into the levels of parameters for previously deposited material which can be learned by the network reducing the locality of the data used for predictions and introducing a weakness during online printing in unseen conditions where the surrounding context will not be systematic and may be out of distribution.

### Bed remover

To reduce the need for human intervention in the printing process and aid continuous printing, a new and simple method for removing completed prints has been developed. Numerous methods have previously been implemented to automatically remove parts upon completion[67,68]; however, previous implementations either require extensive hardware modification, are costly, or only able to remove a relatively limited range of parts. Our bed removal system requires no additional electronics, motors, or complex mechanical parts. The proposed solution can be retrofitted to any extrusion printer and is composed primarily of printed parts which can be produced by the printer in question. The already mobile print head moves and docks with a scraper located to the rear of the build platform. Subsequently, the printer's in-built motors are used to move the print head and scraper across the build surface removing the printed object. After removal, the print head returns the scraper to its home location and undocks (see Supplementary Movie S4). To ensure that the scraper always remains in the same position, a scraper-dock with magnets is attached to the print bed to hold the scraper in place until the next object requires removal. Further details on this system can be found in Supplementary Fig. S3 and Supplementary Note 1 in the Supplementary Information, and also in a GitHub repository containing the modifiable CAD STEP files, STL files for printing, and example G-code scripts for part removal (https://github.com/cam-cambridge/creality-part-remover).

### Training procedure

To train the network we determine the cross-entropy loss at each of the heads and then sum these losses together before backpropagation. This results in the shared backbone of the network being updated to accommodate the loss for each head, with the fully connected layers to each head only being updated by that head's loss. The initial learning rate was selected at each of the 3 training stages by sweeping a range of values and selecting a learning rate with a large drop in loss[69]. Learning rates for each of the stages can be seen in the supplementary information. Selection of the correct learning rate was of key importance—a high learning rate led to poor attention maps, whereas too low learning rates took longer to train or got stuck in early local minima. An AdamW optimiser[70,71] was used during training with a reduce on plateau learning rate scheduler to decrease the learning rate by a factor of 10 when 3 epochs in a row didn't improve the loss by more than 1%. Plots of the learning rate during training can be found in the supplementary information. A training, validation, and test split of 0.7, 0.2 and 0.1, respectively, was used with a batch size of 32. The three stages of training were trained for 50, 65 and 10 epochs, respectively. Each stage was trained three times with three different seeds. During the transfer learning the best seed from the previous stage was chosen as the base to continue training from.

To determine the importance of this multistage training and the use of attention, four different ResNets[47] were trained using the same configuration except only with a single seed and a single stage on the full dataset. The test accuracies can be seen in Table 1 alongside the accuracy of our chosen attention network trained using the three stages. It was found that larger models with these additions actually drop in test accuracy. Applying further pre-training on each model on other datasets would likely improve accuracy across the board.

The use of multiple parameters in a single multi-head network to provide additional context may lead to improved performance over training single parameters in separate networks. After 50 epochs of training, a ResNet18 model with a single head output for predicting flow rate achieved an accuracy of 77.5%. The same model with multiple heads (one for each of the four parameters) after 50 epochs of training achieved a final accuracy of 82.1% at predicting flow rate.

### Computing and software requirements

Final models were trained using half-precision floating-point format (FP16) on two Nvidia Quadro RTX 5000 GPUs with a i9-9900K CPU (eight cores and 16 threads) and 64GB of RAM. This setup was also used for the online correction. Some prototyping work took place on an HPC GPU cluster equipped with Nvidia Tesla P100 GPUs. Neural networks were developed with PyTorch v1.7.1 (https://github.com/pytorch/pytorch), Torchvision v0.8.2 (https://github.com/pytorch/vision), Tensorboard v2.4.1 (https://github.com/tensorflow/tensorboard). Data analysis used Python v3.6.9 (https://www.python.org/), NumPy v1.19.5 (https://github.com/numpy/numpy), Pandas v1.1.5 (https://github.com/pandas-dev/pandas), SciPy v1.5.4 (https://www.scipy.org/), Seaborn v0.11.1 (https://github.com/mwaskom/seaborn), Matplotlib v3.3.3 (https://github.com/matplotlib/matplotlib), Jupyter v1.0.0 (https://jupyter.org/), JupyterLab v.2.2.9 (https://github.com/jupyterlab/jupyterlab) and Pillow v8.1.0 (https://github.com/python-pillow/Pillow). Data collection and parameter correction servers were developed with Flask v1.1.1 (https://github.com/pallets/flask), Flask−SocketIO v5.1.0 (https://github.com/miguelgrinberg/Flask−SocketIO), Octo-Print v1.6.1 (https://octoprint.org/), Marlin 1.1.9 (https://marlinfw.org/). Slicing and print preparation used PrusaSlicer v2.3.3 (https://github.com/prusa3d/PrusaSlicer) and Simplify3D v4.1.2 (https://www.simplify3d.com/).

### Data availability

The labelled image data generated in this study and used to train the model has been deposited in the University of Cambridge data repository (https://doi.org/10.17863/CAM.84082). Source data for plots in figures are provided with this paper. Source data are provided with this paper.

## Code availability

Code used to generate the results in the paper is available in a GitHub repository (https://github.com/cam-cambridge/caxton). Further detail on the bed remover can also be found in a GitHub repository (https://github.com/cam-cambridge/creality-part-remover).

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

## Acknowledgements

This work has been funded by the Engineering and Physical Sciences Research Council, UK PhD. Studentship EP/N509620/1 to D.A.J.B., Royal Society award RGS/R2/192433 to S.W.P., Academy of Medical Sciences award SBF005/1014 to S.W.P., Engineering and Physical Sciences Research Council award EP/V062123/1 to S.W.P. and an Isaac Newton Trust award to S.W.P.

## Author contributions

D.A.J.B. and S.W.P. conceived the idea and designed the experiments. D.A.J.B. developed the data collection pipeline and generated the dataset. D.A.J.B. built, trained and tested the deep learning model. D.A.J.B. designed the control loop and performed experiments. D.A.J.B. generated the model prediction visualisations. S.W.P. provided supervision and guidance during the project. D.A.J.B. wrote the original draft and prepared the figures. D.A.J.B. and S.W.P. contributed to discussions on the experiment and edited and reviewed the manuscript.

## Competing interests

D.A.J.B. is the founder of Matta Labs Ltd., a company in the area of AM error detection. D.A.J.B. and S.W.P. are inventors on a patent submitted by Cambridge Enterprise to the UK Intellectual Property Office (application number: 2204072.9) covering the contents of this report.
