## [Peer Review File · Nature Communications]

Generalisable 3D printing error detection and correction via multi-head neural networksREVIEWER COMMENTS

Reviewer #1 (Remarks to the Author):

This is a solid contribution to the field and one of the best error monitoring systems I have seen. I think it should be published after two major limitations are resolved.

1. The manuscript is missing a lot of the CV work in this area -e.g. all the multiple camera approaches - it would be far stronger to compare the efficacy of this method to all of the previous methods.

2. The CAXTON system used for automated data collection - needs much more detailed description. How does it work? Exactly? How close is it to human labeling? I looked for it on the web - and there is nothing.

These I may have missed - but:

3. Where is the bed remover designs and code?

4. Where is the data set, core code -- I see all the OS components you used - but where is the links to your work and what license are they released under?

Finally, it looks like this approach has solved some of the extrusion based printing errors - however, please discuss how it does with bed adhesion issues and mechanical failure of the printer (e.g. belt slipping).

Reviewer #2 (Remarks to the Author):

Authors have presented a generalisable 3D printing error detection and correction approach using multi-head neural networks. The idea of the paper is interesting; however, the neural networks (NN) foundation of the article in current form has number of major flaws, out of which some are as follows:

1. What is multi-head in NN?

2. How Caxton knows to identify and correct diverse errors? Any mathematical explanation?

3. What is single NN? Do you mean single layered NN? How it self learns interplay between manufacturing process?

4. Collection of image data is not systematic. Why authors choose random parameter selection over systematic parameter selection?

5. 81 classes for prediction using single NN, is not the problem of non-linearity in feature space is going to affect downgrade prediction accuracy?

6. Augmentation of dataset is reasonable, but why synthetic data augmentation is a better choice in rotation case instead of using systematic camera positioning for capturing bigger dataset?

7. Why image-cropping procedure has not been automated?

8. How NN learns inherent bias in the dataset?

It is desirable that the authors should consider above points before moving towards building a prototype for error correction. In the current form the loops holes are very visible in data collection and parameter selection. The reasoning should be strong to make the paper technically sound.

**Point-by-point response to reviewer comments on:
“Generalisable 3D Printing Error Detection and Correction via Multi-Head Neural Networks”.**

Reviewer #1

“This is a solid contribution to the field and one of the best error monitoring systems I have seen. I think it should be published after two major limitations are resolved.”

Comment 1: *“The manuscript is missing a lot of the CV work in this area -e.g. all the multiple camera approaches - it would be far stronger to compare the efficacy of this method to all of the previous methods.”*

We thank the reviewer for pointing this out. We have re-written our discussion of existing CV work and its relative efficacy to include significantly more references and detail, especially on multiple camera approaches. On page 2 beginning line 62, we now discuss 32 CV references, of which 12 are multiple camera approaches and 8 are machine learning implementations.

Comment 2: *“The CAXTON system used for automated data collection - needs much more detailed description. How does it work? Exactly? How close is it to human labeling? I looked for it on the web - and there is nothing.”*

We agree that the specifics of how CAXTON functions were not sufficiently described in the manuscript. We have therefore added almost a new page of description to the methods section to provide considerably more detail on the specifics of how the CAXTON system developed in this work was used for automated data collection (see page 19, line 735). This includes the printer hardware, STL download, randomised toolpath generation, toolpath splitting, image acquisition, parameter sampling, and data storage.

We have also added further detail on page 4 line 188 to describe how the CAXTON system differs from human labelling and how CAXTON is advantageous compared to human labelling both in terms of accuracy and in handling combinations of parameters which are heavily coupled with similar visual features.

It may also not have been clear from original text that CAXTON is a new system developed as part of this report rather than a pre-existing system. Therefore there was no information the reviewer could find on the web. We have made a new public GitHub repository (<https://github.com/cam-cambridge/caxton>) with further details and code for CAXTON to allow the reviewer and community to build on the work. To further clarify this, we have made it more explicit throughout the text which aspects of the work are part of the CAXTON system.

Comment 3: *“Where is the bed remover designs and code?”*

We thank the reviewer for this comment as the bed removal system may be of interest to the community, and thus thorough explanation and design access is needed. We have added an additional figure (figure S3) to the supplementary information showing three exploded views of the bed scraper’s construction, alongside an example G-code sequence for part removal. We have also added a video of the bed remover in operation to the supplementary information (Movie S4) to further illustrate its operation. An additional paragraph has been added to the supplementary information explaining each step in the removal process in detail. We have also created a separate GitHub repository containing the modifiable CAD STEP files, STL files for printing, and example G-code

scripts for part removal (<https://github.com/cam-cambridge/creality-part-remover>). The link to this has been added to the manuscript methods section page 20 line 799.

Comment 4: *“Where is the data set, core code -- I see all the OS components you used - but where is the links to your work and what license are they released under?”*

We have added links to the data set (<https://doi.org/10.17863/CAM.84082>) and core code (<https://github.com/cam-cambridge/caxton>) to the manuscript in the “Code Availability” section (page 22 line 862).

The data set is being hosted on Apollo, the University of Cambridge data repository. There, the data set has a permanent DOI, it is guaranteed to remain unchanged once released, and it will be available indefinitely. This is the first release of a 3D printing imaging data set and we are releasing under the very permissive CC-BY license, which will allow anyone to access and re-use it. The data set will be released on Apollo once this paper has been published. Until this occurs, the data set can be anonymously viewed by the reviewer (and anyone else with the link) on Google Drive (https://drive.google.com/drive/folders/1UUF_lq79Wuj52m_CPoZYoaN9KMJGH6o6?usp=sharing).

The core code is being released under an MIT license, which is a standard permissive license for software and will also allow others to build on our work.

Comment 5: *“Finally, it looks like this approach has solved some of the extrusion based printing errors - however, please discuss how it does with bed adhesion issues and mechanical failure of the printer (e.g. belt slipping).”*

We have added a section to our discussion on the limitations of our method with respect to bed adhesion and mechanical failure of the printer to our discussion (page 18, line 681). In this section we have also further discussed what improvements could be made to the work to address these error types.

Reviewer #2

“Authors have presented a generalisable 3D printing error detection and correction approach using multi-head neural networks. The idea of the paper is interesting; however, the neural networks (NN) foundation of the article in current form has number of major flaws, out of which some are as follows:”

We are pleased that the reviewer thinks the ideas presented in the work are of interest and hope that the revisions we have made regarding the neural network architecture design, dataset acquisition, augmentation, and training process address the concerns raised.

Comment 1: *“What is multi-head in NN?”*

We have now added to the text a detailed description of what a multi-head neural network is, how it differs from the common single head neural network, and the reasoning behind this choice (page 6, line 259 and page 9, line 326). We have also modified the caption for figure 2a detailing the network architecture to describe the network structure more clearly. We thank the reviewer for highlighting this as the added context has greatly clarified the reasoning behind the network architecture selection.

To summarise, 3D printing error correction requires a neural network to simultaneously predict whether the value of each of the four parameters (flow rate, lateral speed, z-offset, and hotend temperature) is either good, high, or low. We use a multi-head neural network to achieve this. In this

network, multiple output heads are used with a shared backbone for feature extraction. The backbone of the network being the first part that interprets the image data input to the network. The weights of the shared backbone are updated during the backward pass in training by a sum of the losses from each of the separate output heads. The training therefore pushes the shared backbone to predict not just one but all four parameters simultaneously. To do this well, the network must learn its own interpretation of the relationships between each of the parameters and the importance of features that are shared across the parameters. The alternative (which is described in more detail in our response to reviewer comment 3 below) would be to use four separate neural networks with single output heads. In this case there would be four separate networks where each backbone only interprets the image data input for a single parameter. Thus the network would not have any information about other parameters, and thus would not be able to learn any interrelationships between them.

Comment 2: *“How Caxton knows to identify and correct diverse errors? Any mathematical explanation?”*

Caxton knows how to identify and correct errors because it has been trained with images labelled with the level of deviation from optimal printing parameters. Thus, when Caxton sees a new image, it can identify how far the printing parameters are from good values, and therefore how the parameters should be changed to reach good values and therefore print well. While this was discussed in the introduction (page 3, line 123), results (page 4, line 168), and methods sections, we agree that the precise workings of Caxton needed more detailed description. As a result, we have added page of additional description to the methods section on page 19 line 735 and released the code for Caxton (<https://github.com/cam-cambridge/caxton>).

Comment 3: *“What is single NN? Do you mean single layered NN? How it self learns interplay between manufacturing process?”*

In our work we have chosen to use a multi-head neural network, as described in our response to the reviewer’s 1st comment. This allows the shared neural network backbone to learn its own interpretation of the relationships between each of the parameters and the importance of certain features shared across the parameters.

An alternative approach would be to use single output neural networks. In this case, there would be four separate neural networks, each with a single output head. Thus, one network would predict whether the flow rate is good, high, or low; another network would predict whether the hotend temperature is good, high, or low; and similarly, there would be separate networks for lateral speed and z-offset. In this case, however, each network only looks at each parameter in isolation and does not receive any information regarding other parameters during training. Therefore, the single output network does not learn about interplay between parameters.

Additionally, a single output neural network approach requires significantly more computational resources during training and in real-world deployment as four separate networks must be trained independently (as opposed to one) and then these networks must be run in parallel during operation.

To clarify this choice for readers, additional information has also been added to the manuscript on page 6, line 259 and page 9 line 326 describing the alternative approach of using multiple neural networks over a single neural network with multiple output heads.

Comment 4: *“Collection of image data is not systematic. Why authors choose random parameter selection over systematic parameter selection?”*

The nozzle-mounted camera sees the material that is currently being deposited by the nozzle, but it also sees other material that has been previously deposited in the periphery of the image. High-performance in error detection and correction requires the network to learn to focus on the material currently being deposited, because this represents the current state of the system. We chose random parameter selection because with random parameter selection there will be no pattern in the previously deposited material that is seen in the periphery of the camera image. Therefore, to accurately predict current manufacturing parameters, the network must focus on the most recently deposited material around the nozzle. Figure 6 suggests that this strategy is reasonably successful as the network focuses on the material around the nozzle.

By contrast, a systematic approach to parameter selection during training may introduce patterns into the printing parameters by which material was previously deposited, and which would then be visible in the periphery of the image. Therefore, the network might be able to correctly predict parameters in the training data not just by looking at the material that was just deposited, but also by looking at the patterns in the periphery of the image. This would reduce the performance of the network and thus its generalisability in two ways. Firstly, it would reduce the ability of the network to focus on the material most recently deposited. Secondly, it would introduce a weakness during printing in objects in industrial practice where previously deposited material will not be printed with systematically varied parameters.

A dedicated paragraph has now been added (page 20, line 772) explaining why a randomised approach to parameter selection was taken over systematic to clarify this to readers.

Comment 5: *“81 classes for prediction using single NN, is not the problem of non-linearity in feature space is going to affect downgrade prediction accuracy?”*

Our network directly predicts 12 classes, as is illustrated in the final layer of the network schematic in figure 2a. There are 12 classes because for each of the 4 printing parameters “Flow rate”, “Lateral speed”, “Z offset”, and “Hotend temp”, a prediction is made as to whether the parameter is “High”, “Good”, and “Low”. Additionally, each parameter can only be one of “High”, “Good” and “Low” at any time. Therefore, there are 81 possible combinations.

Predicting 12 classes is challenging. However, it is the kind of challenge that neural networks excel in. The high dimensionality of the neural network, with many layers and millions of parameters, far exceeds that of the output classes. This is why we selected neural networks for this task. 12 classes is also not an unusually large number for these kinds of algorithms. For example, the ImageNet Large Scale Visual Recognition Challenge (one of the key computer vision benchmarks) uses a trimmed version of the ImageNet dataset with 1,000 classes. The raw ImageNet dataset contains over 20,000 classes.

Non-linearities in the feature space are also challenging, but again this is something that neural networks are good at dealing with. The data within images is high dimensional and non-linear and the neural network backbone serves to reduce this complexity by using the backbone as a feature extractor to map to a lower dimensional latent space representation. This latent or feature space will still be non-linear but the universal approximation theorem suggests that the network can approximate any continuous function - not just linear functions. We have added further detail to our description of the multi-head network on page 6 line 259 and page 9 line 326 to clarify how the network backbone extracts features that enable prediction of the 12 classes.

Finally, despite the challenging number of predictions and data set, we know that the network’s accuracy at predicting all 4 parameters on the unseen test data set was at least 84.3% (and likely higher since for many error types there will be multiple combinations of parameter changes that could correct it). This accuracy is very good and sufficient for the numerous practical demonstrations across geometries, 3D printing geometries and setups in figures 4 and 5.

Comment 6: *“Augmentation of dataset is reasonable, but why synthetic data augmentation is a better choice in rotation case instead of using systematic camera positioning for capturing bigger dataset?”*

Primarily this synthetic approach is more time and resource efficient whilst additionally allowing for a smaller stored raw dataset with augmentations applied at run-time. To clarify this in the manuscript, we now have added reasoning for why the synthetic approach was taken as opposed to generating more raw data through the systematic altering of camera position and environmental lighting conditions (page 6, line 248).

Comment 7: *“Why image-cropping procedure has not been automated?”*

The image-cropping procedure in this work was achieved automatically both during training and testing by using the saved information on the location of the nozzle tip in the image. We thank the reviewer for raising this point as the text did not clearly explain this. We have added that cropping was done automatically throughout the work in the data collection, training, and testing sections (page 4, line 180; page 6, line 241; page 10, line 389).

Comment 8: *“How NN learns inherent bias in the dataset?”*

There are necessarily biases in the data set because it is not feasible to produce a training data set that covers all possible material compositions, colours, printing parameters and printing setups that someone might want to use for extrusion 3D printing. For example, while in our data set the 3D models, slicing settings, and parameter values have been randomly sampled, our data set consists solely of 3D printing with the polymer PLA, printed between 180°C to 230°C, and used the printer’s default limits on feedrate and acceleration in every print. Additionally, because of the large but not infinite size of the data set, there may be certain combinations of parameters which only appear in a specific print or with a single colour of filament and thus the network has learned these incorrect features as mappings. Nevertheless, the network still generalises well to unseen geometries, materials, and setups, (figs 4 and 5) suggesting that it has learnt some features that are universal across extrusion 3D printing (fig. 6).

It is an important issue to highlight and we have therefore added a paragraph discussing possible sources of bias in the data set and how this could affect model training and performance on page 17 line 655. In this paragraph we also highlight potential future work to minimise bias through larger datasets that are more balanced across geometries, materials, parameter combinations, and conditions.

It is desirable that the authors should consider above points before moving towards building a prototype for error correction. In the current form the loops holes are very visible in data collection and parameter selection. The reasoning should be strong to make the paper technically sound.

We thank the reviewer for their helpful comments and believe that additions we have made in response to these comments have strongly clarified the reasoning behind our data collection and parameter selection.

REVIEWERS' COMMENTS

Reviewer #2 (Remarks to the Author):

Unfortunately, the authors failed to respond to major concerns regarding AI in the manuscript reasonably. The answers to almost all queries are given in a manner which can't be deemed as to the point.

For e.g.: "How Caxton knows to identify and correct diverse errors? Any mathematical explanation?", and the answer is "Caxton knows how to identify and correct errors because it has been trained with images labelled with the level of deviation from optimal printing parameters." Another query was "How it self learns interplay between manufacturing process?", and the response by authors is "This allows the shared neural network backbone to learn its own interpretation of the relationships between each of the parameters and the importance of certain features shared across the parameters.", and more.

It is a sincere advice that authors should try to explain the reason (if possible with mathematical justification) behind opting any specific technique, instead of just mentioning that the technique can do wonders. The concept of ML/AI is not of any use if it is presented in terms of a black-box model. Hence, I restate my previous opinion that the paper is quite weak in terms of explaining ML/AI applications.

Unfortunately, the authors failed to respond to major concerns regarding AI in the manuscript reasonably. The answers to almost all queries are given in a manner which can't be deemed as to the point.

For e.g.: "How Caxton knows to identify and correct diverse errors? Any mathematical explanation?", and the answer is "Caxton knows how to identify and correct errors because it has been trained with images labelled with the level of deviation from optimal printing parameters." Another query was "How it self learns interplay between manufacturing process?", and the response by authors is "This allows the shared neural network backbone to learn its own interpretation of the relationships between each of the parameters and the importance of certain features shared across the parameters.", and more.

Response: Neural networks learn to identify things through being trained on data. Our responses describe the data and labels that the network was trained on. A more detailed mathematical explanation of how the neural network learns would be generic to neural networks and is covered by many existing textbooks on neural networks. To help readers who wish to learn more about the fundamentals of neural networks, we have added a reference to page 2 of the manuscript (reference 48).

It is a sincere advice that authors should try to explain the reason (if possible with mathematical justification) behind opting any specific technique, instead of just mentioning that the technique can do wonders.

Response: Our reason for using neural networks for this application is that they have shown better performance in image classification tasks than other algorithms. One of the main reasons for this is described in our previous response: "The data within images is high dimensional and non-linear and the neural network backbone serves to reduce this complexity by using the backbone as a feature extractor to map to a lower dimensional latent space representation. This latent or feature space will still be non-linear but the universal approximation theorem suggests that the network can approximate any continuous function - not just linear functions." Again, a more detailed mathematical explanation of how the neural network learns would be generic to neural networks and is covered by many existing textbooks on neural networks. To help readers who wish to learn more about the fundamentals of neural networks, we have added a reference to page 2 of the manuscript.

The concept of ML/AI is not of any use if it is presented in terms of a black-box model. Hence, I restate my previous opinion that the paper is quite weak in terms of explaining ML/AI applications.

Response: Black-box ML/AI models are already useful in diverse applications including facial recognition, voice assistants, protein folding, and autonomous vehicles. We therefore disagree that black-box ML/AI is not of any use.

Nevertheless, better explainability would improve trust, adoption, and function of many AI/ML approaches. This motivated the explanatory visualisations shown in fig. 6, which is the first time explainability of neural networks has been explored in the context of 3D printing error detection/correction.